# Enhancing Wastewater Depollution: Sustainable Biosorption Using Chemically Modified Chitosan Derivatives for Efficient Removal of Heavy Metals and Dyes

**DOI:** 10.3390/ma17112724

**Published:** 2024-06-03

**Authors:** Jana Ayach, Luminita Duma, Adnan Badran, Akram Hijazi, Agathe Martinez, Mikhael Bechelany, Elias Baydoun, Hussein Hamad

**Affiliations:** 1Research Platform for Environmental Science (PRASE), Doctoral School of Science and Technology, Lebanese University, Beirut P.O. Box 657314, Lebanon; jana.ayach.1@st.ul.edu.lb (J.A.); akram.hijazi@ul.edu.lb (A.H.); hussein.hamad@ul.edu.lb (H.H.); 2CNRS, ICMR UMR 7312, University of Reims Champagne-Ardenne, 51687 Reims, France; agathe.martinez@univ-reims.fr; 3Department of Nutrition, University of Petra, Amman P.O Box 961343, Jordan; abadran@uop.edu.jo; 4Institut Européen des Membranes (IEM), UMR-5635, University of Montpellier, Centre National de la Recherche Scientifique (CNRS), École Nationale Supérieure de Chimie de Montpellier (ENSCM), Place Eugène Bataillon, 34095 Montpellier, France; 5Functional Materials Group, Gulf University for Science and Technology (GUST), Mubarak Al-Abdullah 32093, Kuwait; 6Department of Biology, American University of Beirut, Beirut P.O. Box 110236, Lebanon; eliasbay@aub.edu.lb

**Keywords:** bio-filtration, chitosan-based adsorbent, wastewater treatment, heavy metals, dyes

## Abstract

Driven by concerns over polluted industrial wastewater, particularly heavy metals and dyes, this study explores biosorption using chemically cross-link chitosan derivatives as a sustainable and cost-effective depollution method. Chitosan cross-linking employs either water-soluble polymers and agents like glutaraldehyde or copolymerization of hydrophilic monomers with a cross-linker. Chemical cross-linking of polymers has emerged as a promising approach to enhance the wet-strength properties of materials. The chitosan thus extracted, as powder or gel, was used to adsorb heavy metals (lead (Pb^2+^) and copper (Cu^2+^)) and dyes (methylene blue (MB) and crystal violet (CV)). Extensive analysis of the physicochemical properties of both the powder and hydrogel adsorbents was conducted using a range of analytical techniques, including Fourier transform infrared spectroscopy (FTIR), X-ray diffraction (XRD), Brunauer–Emmett–Teller (BET), and scanning electron microscopy (SEM), as well as ^1^H and ^13^C nuclear magnetic resonance (NMR). To gain a comprehensive understanding of the sorption process, the effect of contact time, pH, concentration, and temperature was investigated. The adsorption capacity of chitosan powder for Cu(II), Pb(II), methylene blue (MB), and crystal violet (CV) was subsequently determined as follows: 99, 75, 98, and 80%, respectively. In addition, the adsorption capacity of chitosan hydrogel for Cu(II), Pb(II), MB, and CV was as follows: 85, 95, 85, and 98%, respectively. The experimental data obtained were analyzed using the Langmuir, Freundlich, and Dubinin–Radushkevich isotherm models. The isotherm study revealed that the adsorption equilibrium is well fitted to the Freundlich isotherm (R^2^ = 0.998), and the sorption capacity of both chitosan powder and hydrogel was found to be exceptionally high (approximately 98%) with the adsorbent favoring multilayer adsorption. Besides, Dubinin has given an indication that the sorption process was dominated by Van der Waals physical forces at all studied temperatures.

## 1. Introduction

Water pollution has become a significant global issue, driven by industrialization and globalization, with projections indicating a worsening situation in the coming years [1]. The presence of various contaminants, including organic and inorganic substances, poses serious environmental and health risks, including carcinogenic effects [2]. Chitosan’s abundance, biodegradability, and inherent chemical functionalities make it a star player in the fight against polluted water. Its powders and hydrogels boast remarkable adsorption capacity for a wide range of pollutants, with tailor-made derivatives targeting specific threats like heavy metals and dyes. This versatility, coupled with the ease of chemical modification, has opened doors for diverse applications. However, challenges remain. Certain pollutants pose a stubborn resistance, demanding further tweaks to chitosan’s armor. Regeneration difficulties and slow adsorption kinetics can also hamper its real-world effectiveness. Additionally, scalability and cost-effectiveness require further optimization for large-scale deployment. Despite these hurdles, the future of chitosan-based pollutant adsorption shines bright. Continuous research and development promise to refine its capabilities, paving the way for a cleaner and healthier planet [3,4,5]. The heavy metals that have recently been linked to water pollution include arsenic, lead, chromium, cadmium, iron, and vanadium [6]. The presence of these metals in water, even in trace amounts, can lead to unimaginable health challenges, such as brain damage, cancer, and systemic interference [7]. Likewise, dyes can build up in wastewater, causing esthetic pollution and damaging photosynthesis. Additionally, consuming methylene blue (MB) can lead to serious health issues [8,9].

Chitin (see Appendix A), a naturally occurring polysaccharide, exhibits slow biodegradability and is commonly found in the exoskeletons of crustaceans such as crabs, shrimp, and crayfish [10]. The build-up of significant amounts of waste material from the processing of crustaceans has become a substantial issue in the seafood industry [11]. Consequently, there arose a necessity to effectively utilize these by-products. Conversely, chitosan (see Appendix A) can be obtained through the process of alkaline deacetylation of chitin. Chitosan, a plentiful and eco-friendly biopolymer, shines with its triple threat**:** biodegradable, biocompatible, and affordable. This adsorption powerhouse, second only to cellulose, boasts active amine and hydroxyl groups that act like magnets for ionic pollutants, making it a valuable tool for cleaner water [12]. Additionally, chitosan is commonly used to clean and purify industrial wastewater [13,14]. In the literature, it has been reported that the amine groups are primarily responsible for adsorption, while the hydroxyl groups aid in stabilizing the interactions between the amine groups and cationic species [15,16].

Chitosan’s ability to grab contaminants hinges on several key aspects: chemical composition, surface area, porosity, and functional groups. The more exposed amine groups (due to high deacetylation), the stronger its grip. Ample surface area, like in nanoparticles or porous structures, provides more “parking space” for pollutants. Open pores make it easier for them to reach the binding sites within. Finally, specific groups like amines and hydroxyls act as chemical magnets, attracting and holding onto targets. Understanding these factors is crucial for crafting optimal chitosan-based solutions for cleaner water [17,18].

Moreover, cross-linking agents, like glutaraldehyde, can be added to chitosan particles to improve their mechanical properties and stability across a broader pH range. Through this process, hydrogels based on chitosan are produced, which are three-dimensional polymeric networks capable of absorbing significant quantities of water or solvents while in their swollen state [19]. These materials have applications in various fields such as pharmacy, medicine and biomedicine, cosmetology, hygiene, and agriculture [20,21,22]. In turn, cross-linked chitosan materials have been widely investigated and used to remove pollutants such as metals, dyes, and phenols from water through adsorption [23,24,25,26].

In this study, chitosan was chemically extracted from shrimp shells, then used as an adsorbent to remove inorganic and organic species from water. The inorganic contaminants used are lead (Pb^2+^) and copper (Cu^2+^) ions, while the organic pollutants are methylene blue (MB) and crystal violet (CV) dyes. The chitosan was extracted in three steps. The initial step in the extraction process involves demineralization and deproteinization to obtain chitin, which is then subjected to deacetylation using a strong base. The adsorbent material was produced as a powder, which was subsequently reacted with a cross-linking agent, glutaraldehyde, yielding a chitosan-based hydrogel. Both the powder and hydrogel forms of the adsorbent were subjected to characterization in order to evaluate their chemical and physical properties. This characterization involved techniques such as Fourier transform infrared spectroscopy (FTIR), nuclear magnetic resonance (NMR), X-ray diffraction (XRD), X-ray photoelectron spectroscopy (XPS), scanning electron microscopy (SEM), zeta potential (ZP), and Brunauer–Emmett–Teller (BET) analysis. The primary objective of our study was to compare the physicochemical characteristics of chitosan powder extracted from shrimp peelings with commercially available chitosan powder and to assess their respective capacities for adsorbing heavy metals and dyes. After concluding that extracted chitosan had many advantages in terms of characteristics and adsorption capacity, the second aim was to synthesize chitosan gel beads. These hydrogels were cross-linked using glutaraldehude agent (GA) to maintain a form and high mechanical gel quality. These chitosan beads were characterized using the same characterization and adsorption techniques as chitosan powder. The ideal form for maximum adsorption (95–98%) of pollutants from industrial water was chitosan in cross-linked hydrogel form.

Then, a batch of adsorption tests was successfully conducted to examine their adsorption removal capacities by varying diverse experimental parameters such as time, pH, concentration, and temperature. In addition, Langmuir and Freundlich adsorption isotherms were modeled at optimum conditions to determine the nature of the interaction of the different studied adsorbates with chitosan. More importantly, successful reusability experiments were performed for the several pollutants used. All the results obtained for the prepared adsorbents were compared to a commercial chitosan powder. The utilization of biofilters in wastewater treatment offers innovations including high removal efficiency, biodegradability, reuse capacity after desorption, and use of natural materials to prevent pollution.

## 2. Materials and Methods

### 2.1. Materials

The shrimp shells were obtained in fresh condition from a seafood restaurant located in Beyrouth, Lebanon. The commercially available chitosan used in this study is a compound with a molecular formula of C_56_H_103_N_9_O_39_ and a minimum purity and deacetylation degree of 75%. It is identified by CAS number 9012-76-4. Glutaraldehyde solution, with a molecular formula of OHC(CH_2_)_3_CHO, was utilized as a 50% concentration solution (CAS number 111-30-8). Hydrochloric acid (HCl) with a concentration of 37% (CAS number 7647-01-0), sodium hydroxide (NaOH) with a minimum purity of 98% (CAS number 1310-73-2), and hydrogen peroxide (H_2_O_2_) with a concentration of 35% (CAS number 7722-84-1) were also employed in the experimental procedures. Lead (II) standard for AAS (Pb^2+^, 1000 mg/L ± 4 mg/L, CAS number 10099-74-8), copper (II) standard for AAS (Cu^2+^,1001 mg/L ± 4 mg/L, 7758-98-7), methylene blue hydrate dye (C_16_H_18_ClN_3_S. × H_2_O, a minimum purity of 97% and CAS number 122965-43-9) were obtained from Sigma-Aldrich (Saint-Quentin-Fallavier, France) for use in the study. Violet crystal dye (C_25_H_30_N_3_Cl, >80%, CAS number 548-62-9) was purchased from HIMEDIA Thane, India.

#### 2.1.1. Shrimp Shells Preparation

The shrimp shells underwent thorough washing with tap water, followed by a final rinse with distilled water. Subsequently, they were placed in a freezer at an extremely low temperature (−70 °C) to preserve freshness for the subsequent steps. After freezing, the shells underwent a specialized process known as lyophilization to eliminate water content. Finally, the shells were finely ground into a powder using a grinder, resulting in particles smaller than 1 mm in size.

#### 2.1.2. Chitosan Extraction Steps

Demineralization: The ground shrimp shell was combined with a hydrochloric acid (HCl) solution with a concentration of 5%. The ratio of shell to solution was maintained at 1:6 (*w*/*v*). The resulting mixture was subjected to continuous stirring for a duration of 24 h at room temperature. After stirring, sodium hydroxide (NaOH) was added to the solution until the pH became neutral, around 7. Subsequently, the mixture was subjected to filtration to isolate the solid material from the liquid phase. The obtained solid material was subsequently transferred to an oven and subjected to a drying process at a controlled temperature of 60 °C for a duration of 24 h. Deproteinization: To extract chitin, the shells with minerals removed were treated with a solution of sodium hydroxide (NaOH) at a concentration of 2N. The ratio of shell to solution was 3:40 (*w*/*v*). The mixture underwent a heating process, reaching a temperature of 100 °C, while being continuously stirred for a duration of 30 min. This step removes proteins and some impurities. After stirring, ortho-phosphoric acid at a concentration of 5% was added to neutralize the solution to a pH of 7. Following that, the solution was filtered to separate the resulting material. The obtained material was then dried in an oven at a precisely controlled temperature of 60 °C until it attained a fully desiccated state. This process resulted in the formation of chitin, a type of polysaccharide. Decolorization: The chitosan, obtained by removing certain elements, was treated with a solution of hydrogen peroxide (H_2_O_2_) diluted to 5%. The ratio used was 1:10 (*w*/*v*). The mixture underwent agitation for a period of 1–3 h, employing a magnetic stirrer for the purpose of thorough mixing and homogenization. The treatment was considered complete when the solution lost its color. Then, the solution was filtered using a special funnel and washed with distilled water to remove any remaining hydrogen peroxide. The resulting chitosan was subjected to a drying process in an oven (50 °C) for 8 h to remove any remaining moisture. Chitosan beads preparation: During the course of the experiment, the chitosan underwent a transformation, resulting in the formation of gel balls. The process involved mixing 1 g of extracted chitosan with 50 mL of acetic acid diluted to 5% at a pH of 4. To remove air bubbles, the mixture was placed in a centrifuge. Then, the solution was slowly injected drop by drop into a NaOH solution with a concentration of 3M using a syringe. The resultant beads were immersed in a NaOH solution at a controlled temperature of 25 °C for a duration of 6 h. Subsequently, the beads were carefully separated through filtration and subjected to rinsing with varying ratios of ethanol and water solutions: 1:9, 3:7, 5:5, and 9:10 [27]. Cross-linking chitosan beads: After optimization of the condition of cross-linking agent, we have chosen a 2% GA with 60 min time of contact between solution and chitosan hydrogels. The chitosan beads were placed in a container, and a precise quantity of 50 mL of GA solutions was added to ensure an even distribution. After undergoing the cross-linking process, the chitosan beads were meticulously transferred into plastic bags. These bags were subsequently placed inside a desiccator, creating a controlled environment free from moisture, thereby preserving the integrity of the beads.

Appendix A on adsorbent characterization is included in the Appendix A.

### 2.2. Structural Characterizations 

The samples’ structures, phase, and crystallinity were determined by scanning electron microscopy (SEM; Hitachi S-4800, Tokyo, Japan). Surface and micropore areas were described using the Brunauer–Emmett–Teller (BET) method, with different data points and relative pressures (P/Po) from 0 to 1. Their structural and crystallinity properties, elemental composition, and oxidation states were analyzed by X-ray diffraction (XRD; PANAlytical Xpert-PRO (Almelo, The Netherlands) diffractometer equipped with an X’celerator detector using Ni-filtered Cu-radiation). Fourier transform infrared spectroscopy (FT-IR) was recorded with the NEXUS instrument (Oakland, CA, USA), equipped with an attenuated total reflection accessory in the frequency range of 400 to 4000 cm^−1^, liquid NMR on a Bruker Avance AVIII-600 NMR (Billerica, MA, USA) spectrometer and solid NMR on a Bruker VNMRS-VARIAN spectrometer, and X-ray photoelectron spectroscopy (XPS) with an ESCALAB 250 spectrometer (Thermo Electron, Waltham, MA, USA; excitation source: Al K*α* monochromatic source, 1486.6 eV), respectively. Granulometric (LA-950V2) methods encompass various techniques to analyze the size distribution of particles and the zeta meter instrument to measure electrophoretic mobility and zeta potential 4.0 Inc.4012.

### 2.3. Adsorption Experiments of Metal Ions

A batch of experiments were carried out for the determination of the optimum sorption of lead and copper metals by commercial, extract, and hydrogel chitosan. The experiments were conducted first with two mother solutions, 1 liter of 5 mg/L lead solution, in addition to 1 liter of 2.5 mg/L of copper solution (both Pb^2+^ and Cu^2+^ were originated from standard solutions (1000 µg/mL)). For each sample, 25 mL of the mother solution was introduced with 0.025 g of extracted powder and hydrogel chitosan, and the sorption was enhanced by a shaker (200 rpm). Different initial concentrations (2–10 mg L^−1^) were evaluated in the adsorption experiments. The initial pH (2–7) was adjusted by adding NaOH (0.1M) and HCl (5%), and the mixture was continuously stirred. The experiments were then undertaken at different times (1–1440 min) and filtered to 0.45 mm. The filtrate was subjected to thorough analysis using the technique of atomic absorption spectroscopy (AAS; PERKIN ELMER spectrophotometer, air–acetylene flame), to determine the final concentration of the metal ion.

The adsorption percentage, *P* (%), was calculated by applying the following equation:(1)$$P\left(\%\right)=\frac{C_{0}-C_{f}}{C_{0}}\times 100$$
where “*C*_0_” and “*C_f_*” represent the initial and final metal ion concentrations in the solution, respectively, measured in milligrams per liter (mg L^−1^).

### 2.4. Adsorption Experiments of Dyes

A series of adsorption experiments were conducted to investigate the adsorption of dyes on chitosan powder and hydrogel. To prepare a stock solution of methylene blue (MB) with a concentration of 1 g/L, the MB powder was dissolved in ultra-pure water in a volumetric flask. Subsequently, daughter solutions with a concentration of 10^−5^ g/L were obtained by diluting this stock solution. A stock solution of crystal violet (CV) with 250 mg/L concentration was prepared by dissolving the CV powder in ultra-pure water in a volumetric flask, and the daughter solutions with concentrations of 25 × 10^−3^ mg/L were obtained by diluting this stock solution. For each sample, 25 mL of the mother solution was introduced with 0.025 g of extracted powder and hydrogel chitosan, and the sorption was enhanced by a shaker (200 rpm). Different initial concentrations (2–10 mg L^−1^) were evaluated in adsorption experiments. The initial pH (2–7) was adjusted by adding NaOH (0.1 M) and HCl (5%), and the mixture was continuously stirred. The experiments were then undertaken at different times (1–1440 min) and filtered to 0.45 mm. The adsorption percentage, P (%), was determined by ultraviolet–visible spectroscopy (UV-2401Pc, UV-VIS recording spectrophotometer SHIMADZU, Kyoto, Japan) by referring to Equation (1). In the equation, “C_0_” and “C_f_” represent the initial and final concentrations of the dyes (MB and CV) in the solution, respectively, measured in milligrams per liter (mg L^−1^).

### 2.5. Physical and Chemical Characterization of the Commercial, Extracted, and Hydrogel Chitosan

Degree of deacetylation. In order to determine the degree of deacetylation (DD), a sample containing 0.2 g of chitosan was mixed with 25 mL of ultra-pure water and 20 mL of hydrochloric acid (0.1 M). The mixture was then stirred for a duration of 30 min. Once the stirring was finished, 25 mL of water was added and stirred again for 30 min. Once finished, 2 drops of methyl orange were added to the solution, which was then titrated with NaOH (0.1 M). The degree of deacetylation was determined by employing the following equation for calculation [28]: (2)$$\mathrm{DD}\;(\%)=2.03\times \frac{v_{2}-v_{1}}{m+0.0042(v_{2}-v_{1})}$$
where “*m*” represents the weight of the sample, “*V*_1_” and “*V*_2_” represent the volumes of 0.1 M sodium hydroxide solution corresponding to the deflection points, and the coefficients “2.03” and “0.0042” account for the molecular weight difference between chitin and chitosan monomer units [29].

NMR experiments. Solution NMR ^1^H, ^13^C, ^1^H-^1^H COSY, ^1^H-^13^C HSQC, and ^1^H-^13^C HMBC NMR spectra were acquired using a Bruker Avance AVIII-600 NMR spectrometer, which was equipped with a 5 mm TCI cryoprobe. The chitosan powder (commercial or extracted) was dissolved in a mix D_2_O-CD_3_COOD 1:1 (*v*/*v*) and heated up to 80 °C to facilitate the solubilization. The solution was then transferred into a 5 mm NMR tube, and the spectra were recorded at 25 °C and 70 °C, respectively. In addition, solid-state ^13^C CPMAS (cross-polarization magic-angle spinning) NMR spectra were recorded on a 300 MHz VNMRS-VARIAN spectrometer. The powder samples were carefully loaded into a 4 mm ZrO_2_ rotor and then rotated at a speed of 12 kHz. A 2.5 ms contact time was used for the cross-polarization transfer, and up to 4000 scans were accumulated. All ^1^H spectra were referenced indirectly to the tetramethylsilane (TMS) ^1^H signal [30].

BET analysis. The determination was made by analyzing the adsorption isotherm of nitrogen gas at a temperature of 77 K. Prior to the analysis, the samples were degassed at 50 °C for 4 h under a vacuum of 10 μm Hg.

Swelling study. This study aimed to analyze how the beads absorbed water, their behavior in acidic and alkaline solutions, and their effect on the adsorption process in treating wastewater. The swelling behavior of the chitosan hydrogel was investigated using gravimetry techniques. The swelling experiments were realized at 37 °C, in buffer solutions at pH 1.2 (0.1 M HCl) and pH 7.2 (KH_2_PO_4_). The swelling ratio, denoted as G, was computed using the subsequent equation:(3)$$G=\frac{W_{s(t)}-W_{0}}{W_{0}}$$
where *W*_0_ represents the initial weight of the hydrogel, *W_s_* represents the weight of the swollen gel, and *t* represents the duration of the swelling process. The swelling kinetics were determined by examining the variation of the swelling ratio, G, over time. Therefore, G can be calculated using Equation (3) [31].

## 3. Results and Discussion

During our experiments, we synthesized chitosan powder and gel extracted from shrimp peelings. Subsequently, we initiated the physico-chemical characterization of the chitosan and conducted a comparative analysis with commercial chitosan. Since the extracted chitosan showed superior characteristics and higher adsorption capacity, this prompted us to synthesize chitosan gel beads. The results demonstrated that the cross-linked chitosan hydrogel had excellent adsorption properties, with a high capacity for removing of Pb^2+^, Cu^2+^, MB, and CV. Lastly, an investigation into isothermal adsorption modelling was conducted. 


*Physicochemical Characterization of Chitosan-Based Adsorbents*


**Morphological Traits.** The particle sizes of extracted and commercial chitosan are shown in Figure 1a1,a2. With a homogenous distribution, a population has been obtained at 451.56 μm and 1754.61 μm for commercial and extracted chitosan (the accuracy guaranteed to within 0.6%), respectively, with a general average distribution of 255.61 for both. Given the similarity in average diameter between the two types of chitosan, it is crucial to consider particle size as a significant factor in adsorption. We can predict that adsorption on both types of chitosan powder will be similarly effective [32]. As the chitosan gel is in solid form, we were unable to characterize it using this type of technique. To further signalize the morphology of chitosan, the SEM was used to characterize and compare the surface of different particles, including shrimp shell particles (as shown in Figure 1c1), extracted chitosan particles (as shown in Figure 1c2,c3), commercial chitosan particles (as shown in Figure 1c4), and chitosan beads with and without a cross-linking agent (as shown in Figure 1c5). The SEM images showed that shrimp shell particles had a smooth surface with some rough, irregular membrane fissures. Extracted chitosan particles had isolated smooth areas with several pores, indicating their formation [33]. Commercial chitosan particles had a smooth surface with some agglomeration and no pores. According to SEM images, cross-linked chitosan beads had a rough and wrinkled surface, while chitosan beads without the cross-linking agent were relatively smooth. Later, this was confirmed by XRD diffractogram, that the cross-linked gel had no crystal peak, indicating that it is an amorphous material [34]. The size and morphology of the commercial, extracted, and hydrogel chitosan particles obtained in this study were found to be comparable to those reported in the literature. Muley et al.’s SEM analysis revealed that the chitosan particles exhibited a spherical to globular shape, with an average particle size of 92.84 μm found to be considerably smaller than that observed in our chitosan. Moreover, the particles exhibited a rough surface texture characterized by small pores [35]. Ramachandran et al. show that the chitosan particles linked with glutaraldehyde exhibited a smooth surface and a narrow particle size distribution ranging from 105 to 219 μm. Furthermore, the scanning electron microscopy (SEM) analysis revealed that these particles appeared to be fairly smooth and spherical in shape, with a uniformly homogeneous surface [36].

Then, the extracted and commercial chitosan powders were inspected using the BET technique to confirm the SEM image indications. As the chitosan gel is in solid form, we were unable to characterize it using the BET technique. The N_2_ adsorption–desorption findings (see Table 1) demonstrate a notable enhancement in the surface area, pore volume, and pore size of the extracted chitosan (7.43 m^2^/g) in comparison with the commercial chitosan (0.27 m^2^/g). The observed morphology by SEM of the extracted chitosan confirms the presence of significant pores at the surface, which are not present in the commercial case [37,38], while the disparity in nitrogen adsorption (0.27 m^2^/g vs. 7.43 m^2^/g) suggests incomplete deacetylation during extraction of the chitosan. Residual acetyl groups could hinder pore formation, explaining the lower surface area. However, nitrogen-containing impurities or differing chitin sources remain possibilities. To determine the definitive cause, we analyze for residual acetyl groups, identify any impurities, and compare the chitin sources used for each sample. This results are commonly observed in mesoporous materials, of which the sharp increase in the adsorbed quantity at high relative pressure indicates that the available mesoporous volume is occupied [39]. Figure 1b1–b3 shows the purity phases of the commercial, extracted, and hydrogel chitosan obtained by XRD. Both types of chitosan are recognized for their similar semi-crystalline structure, which shows two main distinct diffraction peaks at 2θ angles, 10.82° and 20.19°, that correspond to the plan of (020) and (110), respectively [40]. The XRD analysis of chitosan reveals the characteristics of an amorphous polymer. This indicates a higher degree of disorder in the alignment of polymer chains in the nanoparticles after cross-linking [41]. An X-ray analysis of different chitosan samples (extract, gel, and commercial) revealed contrasting structures. X-ray diffraction analysis revealed peaks at 30° in the lab-prepared samples (extract and gel), suggesting the presence of amorphous regions (disorganized chitosan molecules). Conversely, the commercial chitosan lacked this peak, suggesting a mostly crystalline structure with minimal amorphous regions. This highlights potential differences in the organization of chitosan depending on its source and processing. Taking into account the broadening of each peak in XRD, mean crystallite size has been calculated using Scherrer’s equation in that the approximate crystallite sizes of commercial, extracted, and hydrogel chitosan samples are about 47 nm [42]. Table 1 provides the elemental surface composition of both commercial and extracted chitosan. Appendix A illustrates representative peaks for O 1s, N 1s, and C 1s. The C 1s peak exhibits three distinct components: one at 284.8 eV, characteristic of carbon solely bound to carbon and hydrogen [C-(C,H)]; another near 286.2 eV, indicating carbon forming a single bond with oxygen or nitrogen [C-(O,N)]; and a component near 287.7 eV, indicative of acetal and amide bonds [O-C-O, N-C=O] [43,44]. The significant contribution to the N 1s peak, observed at approximately 399.3 eV, originates from non-protonated amine or amide groups, while a minor component at 401.3 eV suggests the presence of protonated amine. Additionally, the main contribution to the O 1s peak, located at 532.7 ± 0.1 eV, arises from the oxygen in the polysaccharide backbone, while a smaller component near 531.0 eV may be attributed to amide groups resulting from acetylated functions.

**FTIR analysis.** In addition to the morphological traits, it is important to discuss the functional group in the surface of each chitosan. The surface characterization, which was initially performed by SEM and BET, is now being followed by IR analysis to investigate the primary functional groups present on the surface. Figure 2 displays the bands observed in the FTIR spectra of the commercial, extracted, and hydrogel chitosan samples. The spectra exhibited several notable peaks. A peak at 3418.21 cm^−1^ indicated the symmetric stretching vibration of O-H and amine. Additionally, an absorption peak at 2922.59 cm^−1^ suggested the presence of C-H stretch. The absorption band at 1639.2 cm^−1^ was attributed to C=O stretching (amide I), while peaks at 1155.15 cm^−1^ and 1076.08 cm^−1^ represented C-O stretching. The spectra of commercial chitosan and extracted chitosan were compared, revealing similar stretching vibrations with slight shifts. These findings suggest a strong structural similarity between the extracted and commercial chitosan samples. FTIR analysis confirms the successful reaction of amino groups in chitosan and the absence of free amino groups in the cross-linked chitosan incorporated in scaffolds [45]. The presence of imine bonds in the scaffold spectra signifies the successful cross-linking process. The presence of various chemical groups, including C-H and C-OH stretches, suggests the formation of carbon–hydrogen bonds. The increasing peak areas in the 1653–1664 cm^−1^ range indicate a higher degree of cross-linking driven by the Schiff base reaction involving carbonyl groups. Overall, these findings demonstrate the effectiveness of the cross-linking process in chitosan-based scaffolds [46]. The FTIR spectrum of chitosan powder provides important information on its molecular structure and functional groups, aiding in identifying specific groups, confirming the degree of deacetylation (83.88%), determining structural conformation, and monitoring chemical modifications. 

**Nuclear Magnetic Resonance.** In addition to the IR spectrum, the ^1^H and ^13^C NMR spectra of commercial and extracted chitosan are shown in Figure 3. The assignment, confirmed by the ^1^H–^1^H COSY, ^1^H–^13^C HMBC, and ^1^H–^13^C HSQC 2D spectra, is displayed on top of the ^1^H and ^13^C (Figure 3b,d) spectra of synthesized chitosan and is in agreement with the literature [47]. The intense peaks at ~2/20 ppm (^1^H/^13^C) are assigned to the residual proton and ^13^CH_3_ of the acetyl group. The lack of signals characteristic of carbonyl (C=O) and methyl (-CH_3_) carbons of the acetamide group in the ^13^C NMR spectrum of the extracted chitosan, solubilized in deuterated acetic acid, suggests that the deacetylation degree (DD) is important in the solubilized fraction, in agreement with the calculation of 83.88% presented above. The appearance of a small methyl signal in the commercial solubilized chitosan indicates a lower DD in this sample compared to the extracted chitosan. 

The solid-state ^13^C NMR CPMAS spectrum (see Figure 4a,b) shows a small peak at ~23 ppm, indicating the small contribution of the residual methyl group in the acetamide moiety of chitosan [48]. A small peak is also visible at the carbonyl chemical shift. Collectively, these observations on the powdered extracted chitosan provide evidence of the efficiency of the deacetylation step. The C1 ring signal appears at ~102–105 ppm, the C2 ring signal shifted to ~55–57 ppm owing to the attached amino group, and the signal at ~60 ppm is relevant to C6 in the ring [49]. Peaks in the region of 73 and 80–85 ppm correspond to the C3, C4, and C5 signals, in agreement with the assignment of the solubilized sample and with the literature for the powder chitosan [48,50,51,52]. Furthermore, the confirmation of the C=N bond formation between chitosan and glutaraldehyde was achieved by comparing the ^13^C CPMAS spectrum of this sample with those obtained from the control samples of commercial and extracted chitosan powders (see Figure 4c). The presence of C=N bonds in chitosan enhances its mechanical strength and makes it well suited for further applications as a cellulose matrix [53]. This characteristic can improve the chitosan’s capacity for removing heavy metals by increasing its hydrophilicity and creating a microporous structure [54]. Furthermore, the obtained ^13^C CPMAS results were in agreement with the FT-IR and XPS analyses by placing emphasis on the existence of carbonyl and amine functional groups [55]. 

**Swelling study and zeta potential.** Figure 5 shows the swelling ratio G for each data point of the swelling curve. The swelling study was conducted for chitosan cross-linked gel since this study is only undertaken for samples in solid form with good mechanical characteristics. The swelling ratio of the material exhibited a substantial decrease upon adjusting the pH from 1.2 to 7.2. Moreover, the equilibrium swelling was achieved much faster at pH 7.2 (within 5 h) compared to pH 1.2 (approximately 20 h). Chitosan hydrogels exhibit a sweet spot for swelling at acidic pHs (below 4) due to strong amino group protonation, but biocompatibility concerns arise. Neutral pH (around 7) sees moderate, biologically relevant swelling, while basic pH (above 9) leads to minimal swelling due to deprotonation and potential controlled release applications. Consider deacetylation degree, cross-linking, and ionic strength for further nuance. The observed alterations in the swelling behavior can be attributed to the protonation of the amine groups in chitosan at a pH of 1.2. This protonation process results in electrostatic repulsion between the polymer segments and generates osmotic pressure within the network, facilitated by the presence of counter ions. Furthermore, the ionization of the amine groups may lead to the disruption of hydrogen bonds between chitosan molecules, thereby promoting relaxation of the macromolecular chain. In contrast, at pH 7.2, the absence of amino group ionization significantly reduces swelling [31]. In order to examine the adsorption behavior of metals and dyes on chitosan adsorbent at different pH levels, a ZP analysis was conducted. The swelling study of the chitosan samples was found to have a direct correlation with their zeta potential values. The zeta potential of both adsorbents (commercial and extracted chitosan) exhibits positive values at pH range (2–6), and it decreases to a negative value at pH range as the pH increases (6–9). The highly acidic surface of both solids can improve their inability to bind to positively charged ions in solution (metals). At pH 6, the isoelectric point has been proven to be the favored capacity of adsorption at this part. From pH 7, the zeta potential becomes highly negative, and this can mean the precipitation of metals in the solution and, therefore, the formation of colloidal commercial and extracted adsorbents [30].

Adsorption experiments were conducted using lead and copper ions, as well as methylene blue and crystal violet dyes.

Chitosan in commercial/extracted powder form and cross-linked gel, according to the conditions already valued in the parts below, were used for the adsorption of metals and dyes.

**Effect of contact time.** The data obtained from the adsorption of Pb^2+^, Cu^2+^ ions, MB, and CV dyes onto chitosan revealed that the adsorption increased with longer contact time. Figure 6 illustrates the uptake of Pb^2+^ and Cu^2+^ ions by commercial chitosan during the initial 1 min of agitation; a slower sorption rate was observed initially, with equilibrium being reached within 30–60 min. Subsequent increases in contact time had minimal effect on the amount of ions adsorbed. The total sorption of lead (Pb^2+^) and copper (Cu^2+^) solutions, with initial concentrations of 5 mg/L and 2.5 mg/L, respectively, was examined, respectively, and resulted in a metal uptake of 74.41% and 98.98%, respectively. Similarly, the uptake of MB and CV by commercial chitosan showed rapid sorption during the first 10 min of agitation, followed by a slower rate and equilibrium achieved after 120 min. The total sorption of MB and CV solutions with initial concentrations of 10–5 mg/L each resulted in a metal uptake of 98% and 80%, respectively. The initial fast uptake can be attributed to the availability of free binding sites on chitosan, allowing for quick binding of metal ions and dyes. As the binding sites became saturated, the uptake rate slowed down due to competition for limited active sites by metal ions and dyes [56,57,58]. After analyzing the test results, it was determined that an agitation time of 120 min would be maintained for the remaining batch experiments in order to ensure that equilibrium was achieved. Furthermore, the adsorption capacity of both ions on the extracted chitosan is slightly higher than the adsorption data obtained on the commercial chitosan. This may be due to the higher accessibility to the extracted chitosan pores compared to the commercial one.

**Effect of pH.** The adsorption of heavy metal ions and dyes is significantly influenced by the pH of the solution. This is primarily due to its direct impact on the surface charge of the adsorbent and the specific types of metal ions that are present. Appendix A illustrates the effect of pH on the adsorption of Pb^2+^, Cu^2+^, MB, and CV by commercial, extracted, and hydrogel chitosan. The highest removal efficiencies for lead (Pb^2+^) and copper (Cu^2+^) ions are observed at pH 7, with 85% and 81% removal efficiencies, respectively. This demonstrates that as the pH level becomes more basic, the substance’s ability to adsorb increases. Regarding MB, the maximum removal efficiency of 99% is attained at pH 5, while for CV, it is 99.8% at pH 4 for commercial chitosan. At low pH values, the concentration of H^+^ ions are high, leading to proton competition with metal ions for surface sites. This confirms the low adsorption capacity at lower pH [59]. Metal adsorption is more pronounced at higher pH values because there are fewer protons, more negatively charged ligands on the surface, and reduced competition between H^+^ and metal cations [60]. At pH values above 5, in addition to adsorption, the precipitation of insoluble lead and copper hydroxide salts also takes place in the solution [61].

**Effect of Temperature.** The impact of temperature on the adsorption process of lead (Pb^2+^), copper (Cu^2+^), methylene blue (MB), and crystal violet (CV) was examined at various temperatures (25 °C, 35 °C, 45 °C, and 55 °C) while maintaining a pH of 6 for metals and 5 for dyes. The initial concentrations of metals were 5 mg/L for Pb^2+^ and 2.5 mg/L for Cu^2+^, and for dyes, it was 10–5 mg/L for both MB and CV. The purpose of studying the effect of temperature was to gain a deeper understanding of the adsorption process, and the findings are presented in Appendix A. The maximum adsorption capacity for Pb^2+^ was observed at 55 °C, resulting in an 84% removal rate. Conversely, Cu^2+^ exhibited the highest adsorption capacity of 96.4% at 25 °C. Similarly, for MB, the highest adsorption capacity of 99.93% was achieved at 25 °C, while for CV, it was 99.92% at 55 °C, all using commercial chitosan as the adsorbent. The removal rates of Pb^2+^ and MB were found to increase with higher solution temperatures, suggesting that the adsorption processes are characterized as endothermic . On the other hand, the rate of Cu^2+^ and CV removal improved with lower solution temperatures, suggesting that the adsorption processes are exothermic [62], and this can be explained by the fact that increasing the temperature provides the adsorbed species with enough energy to be released from the surface. Consequently, the desorption rate surpasses the adsorption rate, leading to a higher rate of desorption [59]. 

**Effect of concentration.** The adsorption percentages of commercial, extracted, and hydrogel chitosan in removing of Pb^2+^, Cu^2+^, MB, and CV from water at different initial concentrations are presented in Appendix A. It has been noticed that as the concentration of metal ions in the water increases, the removal percentage also increases. However, after reaching a certain concentration level, the removal percentage levels off. This is because there are more metal ions and dyes available for adsorption, but the amount of adsorbent remains constant. As the concentration of metal ions and dyes increases, more binding sites on the adsorbent are occupied, leading to higher adsorption capacities [60]. Afterward, due to the increased concentration gradient and higher number of ions being adsorbed per unit of the adsorbent, the adsorbent reaches a point of saturation. As a result, the ions are not adsorbed and remain free in the solution.

**Comparison between the uptake of metals and dyes on chitosan.** Lead’s and copper’s differing adsorption onto chitosan stems from a mix of factors. Copper’s smaller size and higher charge and chitosan’s inherent selectivity favor its adsorption. Competition from other ions, solution pH, and chitosan properties like deacetylation degree and pore size further influence their individual adsorption percentages, highlighting the need for tailored experimental conditions to optimize metal uptake.

The differing adsorption of methylene blue (MB) and crystal violet (CV) onto chitosan is a complex dance. MB’s larger size and potential for multi-site binding favor its uptake, while CV’s concentrated charge offers stronger electrostatic attraction. Competition from other ions, pH-dependent ionization, and even chitosan’s own structure (deacetylation and pore size) further influence their individual adsorption behaviors. Understanding these intricate interactions and tailoring experimental conditions are key to unlocking optimal dye-removal efficiency.

**Adsorption isotherm models.** Using the optimization of several factors in the adsorption of metals and dyes by chitosan powder and beads, an isotherm modeling study was conducted to obtain valuable insights into the mechanisms underlying adsorption processes. This information is crucial for the design of effective adsorption systems.

Langmuir isotherm. The adsorption data were analyzed using the linear form of the Langmuir isotherm. However, the nonlinear plots of specific sorption Ce/qe against the equilibrium concentration Ce for MB and CV suggest that the Langmuir principle does not apply in this case. This is indicated by a weak correlation (R^2^ < 0.7) [63,64,65]. This indicates that the adsorption on the solid surface does not occur solely as a monolayer. This could imply the presence of multiple layers of adsorbate molecules on the surface or the influence of other factors that the Langmuir model fails to consider in the adsorption process [57].

Freundlich isotherm. The Freundlich isotherms of the adsorption of Pb^2+^ and Cu^2+^ on the commercial and extracted chitosan particles are shown in Appendix A. Table 2 presents the Freundlich isotherm constants n and their corresponding correlation coefficients R^2^. The adsorption of lead and copper ions onto the various adsorbents resulted in a linear relationship. Values of ‘n’ ranging from 0 to 10 indicate favorable adsorption [66,67,68]. However, the adsorption of Cu^2+^, Pb^2+^, MB, and CV on extracted chitosan particles is the most favored, rather than on commercial chitosan. The Freundlich isotherm is commonly used to describe adsorption on heterogeneous surfaces with a uniform energy distribution. In the literature, it has been observed that the Freundlich isotherm can be divided into multiple sections, sometimes up to five. This division is often attributed to irregular energy distributions caused by different surface groups having varying activation energies for different adsorption reactions. In our study, the Freundlich analysis revealed three distinct linear regions, which can be attributed to the adsorption onto different surface groups, such as amino, acetyl, or hydroxyl groups, or purely physical sorption [69]. Moreover, the correlation coefficients (R^2^) obtained from the Langmuir and Freundlich isotherm models indicate that the Freundlich model is more suitable for describing the adsorption of Pb^2+^, Cu^2+^, MB, and CV on both commercial and extracted chitosan. This preference can be attributed to the initial saturation of sites due to the stronger adsorption of metals and dyes. As the occupation of adsorbent sites increases, the adsorption strength decreases [59].

Dubinin–Radushkevich isotherm. The determination of the mean adsorption energy (E) through the D-R isotherm calculation offers valuable insights into the physicochemical properties associated with the adsorption process. For E < 8 kJ·mol^−1^, physisorption dominates the sorption mechanism [70]. When E falls within the range of 8 to 16 kJ/mol, ion exchange becomes the dominant factor in the sorption process. However, if E exceeds 16 kJ/mol, sorption is predominantly controlled by particle diffusion. In this study, the adsorption energies are less than 2 kJ·mol^−1^, suggesting that the sorption process was dominated by physical forces at all studied temperatures (see Table 3) [70]. 


**A comparison of the chitosan removal capacity between previous studies and our experimental results.**


According to the Table 3, the chitosan extracted from shrimp peelings and subjected to cross-linking exhibited remarkably high percentages of heavy metal and colorant adsorption (approximately 85–99%). These values significantly surpass the results obtained in previous studies (see Table 4), where adsorption percentages for copper were below 10%, for lead were around 90%, for CV were below 10, and for MB were a high percentage (96%). Furthermore, even with modifications or the addition of other compounds, other studies failed to achieve the same level of adsorption as our results. Our extraction method, which is simple, cost-effective, and environmentally friendly, yielded a chitosan extract with competitive adsorption percentages for heavy metals and dyes compared to current research.

A novel method for wastewater treatment uses chitosan extracted from shrimp shells. This method is environmentally friendly, reduces waste generation, and utilizes a customizable hydrogel chitosan for effective treatment. Furthermore, the performance of chitosan-based hydrogels in heavy metal and dye adsorption stands out with remarkable results. These hydrogels exhibit a high adsorption capacity, reaching approximately 98%, surpassing many previous studies. Additionally, the efficiency of chitosan hydrogels in adsorbing pollutants is achieved with a minimum number of passes (1–2 passes), providing a cost-effective and time-saving solution. The excellent characterization of the morphology of chitosan beads in our study reveals their remarkable properties, including high porosity and strong mechanical strength. These features contribute to the longevity of the beads (2 years from the beginning of the experimental study), allowing for their repeated use without degradation. The high porosity of the beads (SEM images in Figure 1) enables efficient mass transfer and enhanced adsorption capacity, while their strong mechanical properties ensure durability and resistance to physical stress during application. Combining strong morphology with reusability, chitosan beads offer long-term effectiveness and economic benefits for wastewater treatment. Additionally, chitosan’s natural origin, waste reduction, innovative synthesis, and high adsorption capacity highlight its potential as a sustainable and efficient solution for wastewater management.

## 4. Conclusions

Chitosan was extracted from shrimp shells and characterized by FT-IR, elemental analysis, NMR, XRD, SEM, BET, ZP, particle size, and swelling studies. The extracted chitosan displayed unique properties compared to commercial chitosan, including a more homogeneous surface and larger particle size distribution. Then FT-IR confirmed the presence of key functional groups in both extracted and commercial chitosan, suggesting structural similarity. NMR analysis further validated the successful deacetylation of extracted chitosan and the presence of C=N bonds, indicating enhanced mechanical strength. In addition, the ^1^H and ^13^C NMR spectra of commercial and extracted chitosan were compared, confirming the successful deacetylation of the extracted chitosan. Based on the degree of deacetylation, which was determined to be 83.88%, the material can be classified as chitosan. The extracted chitosan has a homogeneous surface with an average of 255 µm of distribution particle size. The extracted chitosan demonstrated exceptional adsorption capacity for Pb^2+^, Cu^2+^, MB, and CV, exceeding the performance of commercial chitosan. This characteristic increases the removal capacities of Pb^2+^, Cu^2+^, MB, and CV with percentages of 99%, 75%, 98%, and 80%, respectively. These percentages were much higher compared to those of commercial chitosan, which were 70%, 65%, 85%, and 70%, respectively. In addition, the adsorption capacity of chitosan hydrogel cross-linked with GA for Cu^2+^, Pb^2+^, MB, and CV was equal to 85%, 95%, 85%, and 98%, respectively. In addition, Langmuir and Freundlich isotherms were applied to evaluate the adsorption process. While Langmuir did not fully fit the data, Freundlich indicated favorable adsorption with chitosan particles, showing better performance than cross-linked gel particles. Dubinin–Radushkevich isotherm analysis suggested physical forces as the primary driving mechanism. Finally, chitosan-based materials, particularly extracted chitosan, show promising potential for wastewater treatment due to their high adsorption capacity for heavy metals and dyes. Further research is needed to optimize their performance, explore combinations with other materials/technologies, and address challenges for practical implementation. This will contribute to sustainable water treatment solutions and alleviate water pollution concerns.

## Figures and Tables

**Figure 1 materials-17-02724-f001:**
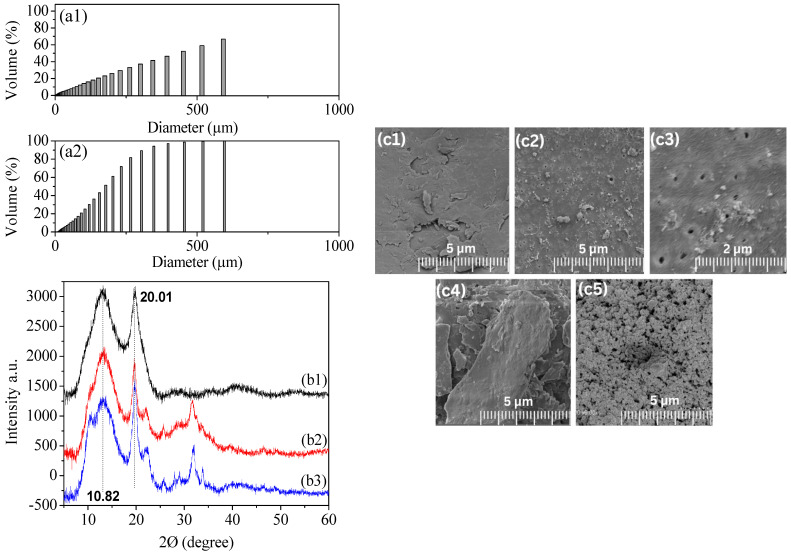
Particle size distribution by granulometry of extracted chitosan (**a1**) and commercial chitosan (**a2**). The XRD spectrum of commercial (**b1**), extracted (**b2**), and hydrogel chitosan (**b3**). SEM images of commercial chitosan (**c1**), extracted chitosan (**c2**), extracted chitosan pores (**c3**), shrimp shells powder (**c4**), and chitosan hydrogel cross-linking (**c5**).

**Figure 2 materials-17-02724-f002:**
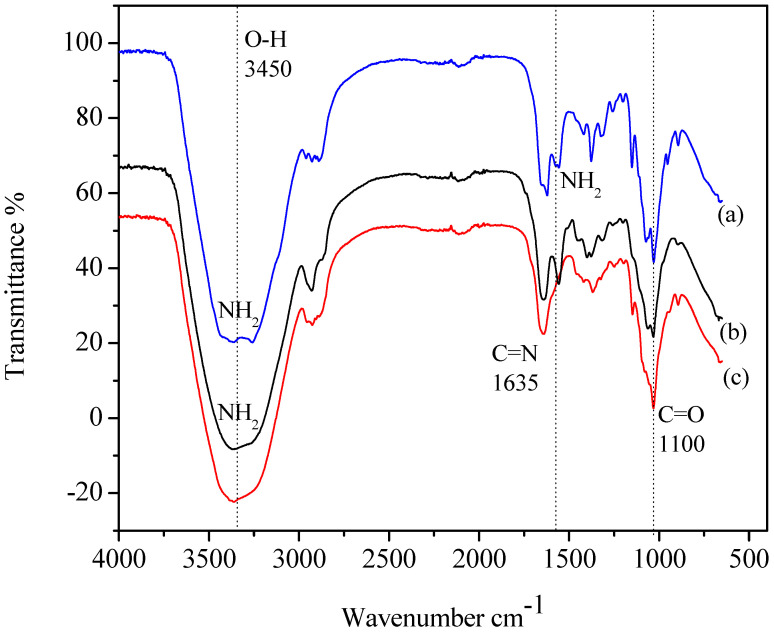
FTIR spectra of extracted chitosan (**a**), commercial chitosan (**b**), and hydrogel chitosan (**c**).

**Figure 3 materials-17-02724-f003:**
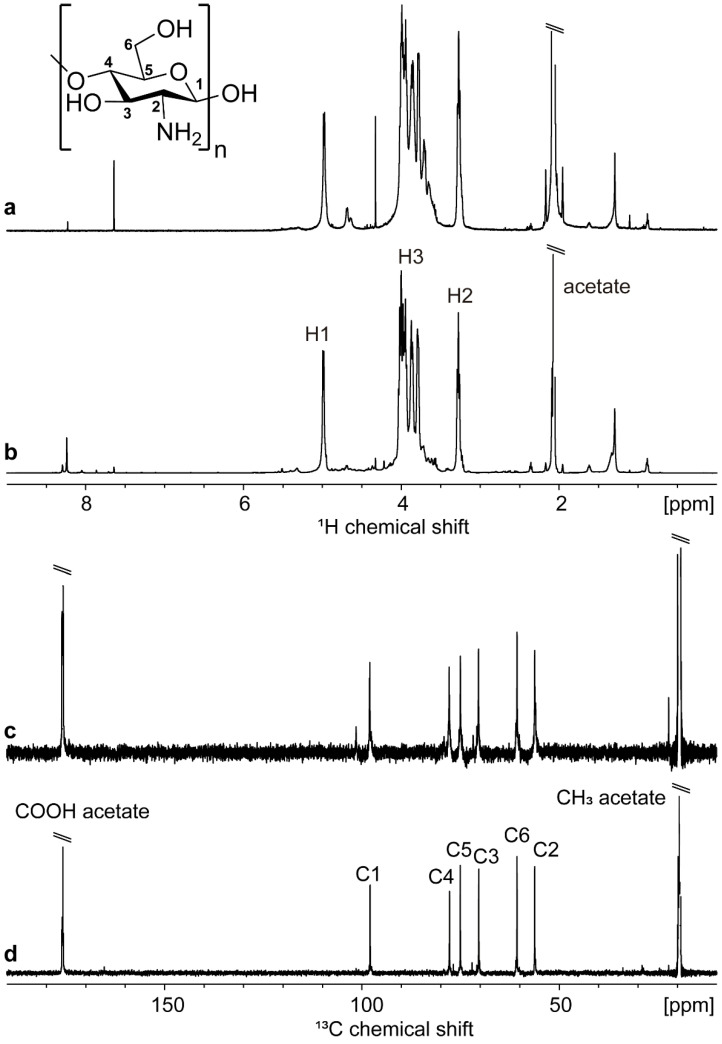
^1^H and ^13^C NMR spectra at 343 K of commercial (**a**,**c**) and extracted (**b**,**d**) chitosan in D_2_O:CD_3_COOD 1:1 (*v*/*v*) with the corresponding assignment.

**Figure 4 materials-17-02724-f004:**
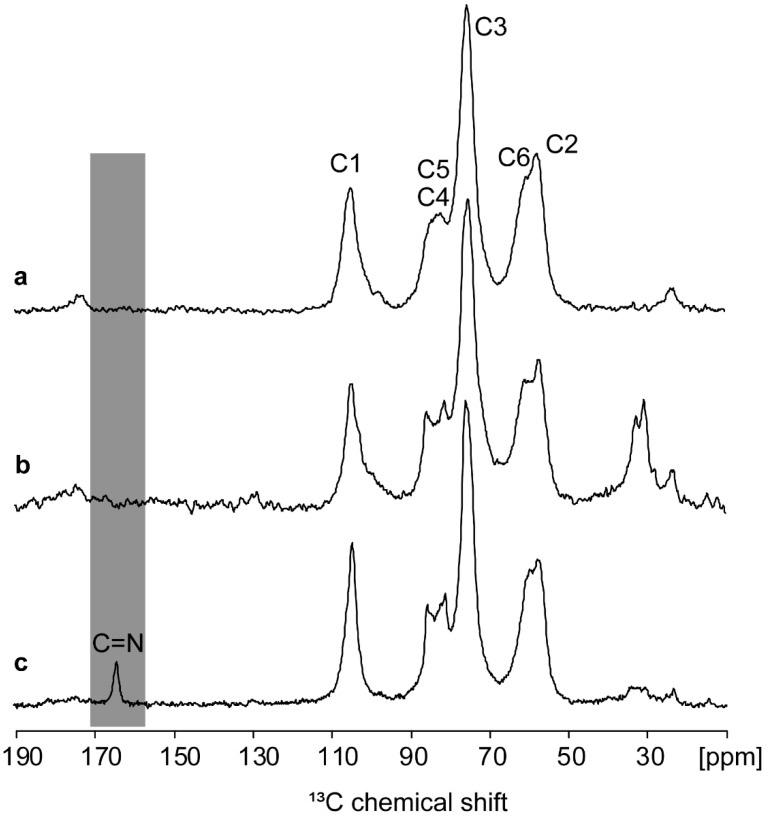
^13^C CPMAS spectra at 12 kHz rotation on a 300 MHz (^1^H Larmor frequency) magnet spinning for extracted (**a**), commercial (**b**), and hydrogel (**c**) chitosan samples and their corresponding assignment. At this spinning frequency, all spinning sidebands are outside the displayed spectral region. The formation of the cross-linked chitosan is confirmed by the appearance of the characteristic C=N resonance in the hydrogel (see grey region).

**Figure 5 materials-17-02724-f005:**
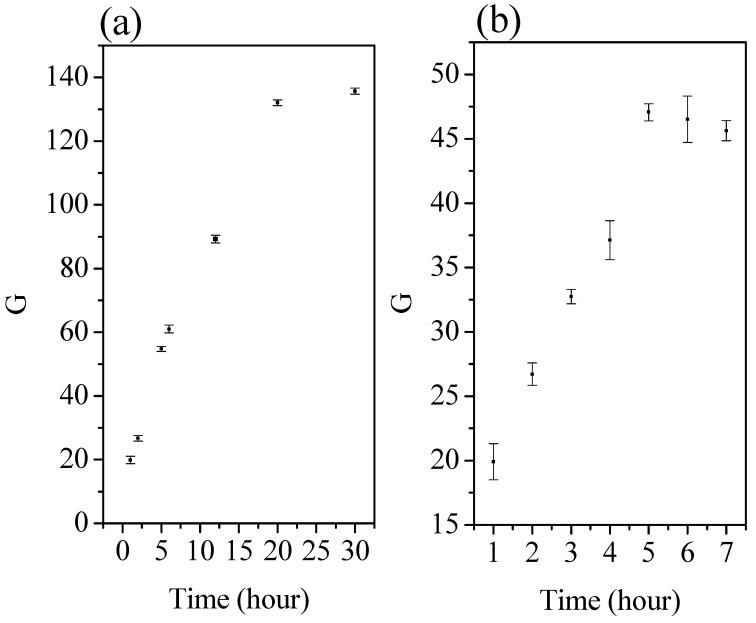
Swelling study of chitosan hydrogel at acidic medium (pH 1.2) (**a**) and the swelling study of chitosan hydrogel at very mildly basic medium (pH 7.2) (**b**).

**Figure 6 materials-17-02724-f006:**
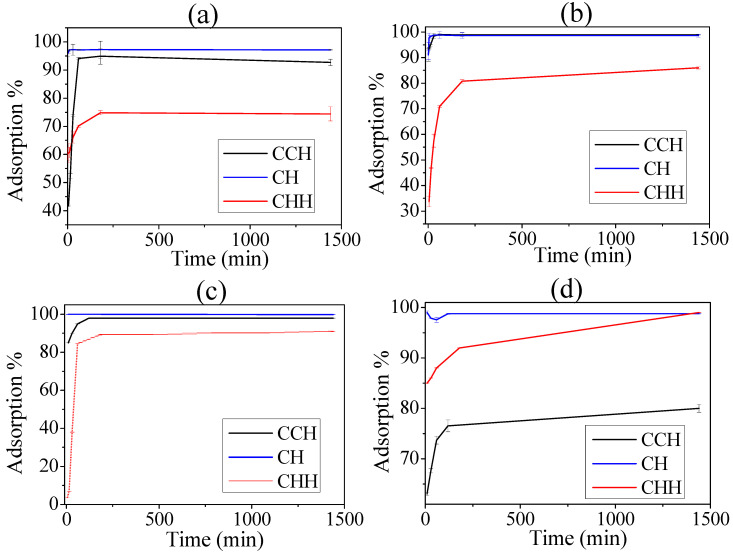
Adsorption of heavy metals and dyes in function of time of contact; adsorption of lead in f(time) (**a**); adsorption of copper in f(time) (**b**); adsorption of MB in f(time) (**c**); adsorption f CV in f(time) (**d**) with the three samples, commercial chitosan (CCH), extracted chitosan (CH), and hydrogel chitosan (CHH).

**Table 1 materials-17-02724-t001:** The physical and chemical characterization results of both the commercial and extracted chitosan samples.

Parameter		Commercial Chitosan	Extracted Chitosan
BET surface area (m^2^/g)		0.27	7.43
Total pore volume (cm^3^/g)		0.01	0.021
Average pore diameter (nm)		7.02	9.46
Elemental analysis (XPS in %)	O 1s	35.31	29.08
	N 1s	3.50	6.65
	C 1s	61.19	64.27

**Table 2 materials-17-02724-t002:** Summary table of heavy metal and dye adsorption percentages as a function of the factors studied.

Factors		Chitosan Commercial				Chitosan Extracted				Chitosan Hydrogel			
		Pb^2+^	Cu^2+^	MB	CV	Pb^2+^	Cu^2+^	MB	CV	Pb^2+^	Cu^2+^	MB	CV
1	Time of contact (min)	60	60	120	120	60	60	120	120	60	60	120	120
	Maximum Ads %	93	94	95	76	98	99	99	98	75	80	83	95
2	pH	7	7	4.5	4.5	7	7	4.5	4.5	7	7	4.5	4.5
	Maximum Ads %	60	75	99	95	98	80	95	96	92	99	85	99
3	Temperature	40	25	40	25	40	25	40	25	40	25	40	25
	Maximum Ads %	82	94	99	98	95	96	99	90	95	98	83	97
4	Concentration	8	4	10	2	8	4	10	2	8	4	10	2
	Maximum Ads %	80	92	99	98	80	96	99	98	85	96	92	98

**Table 3 materials-17-02724-t003:** Freundlich and Dubinin–Radushkevich study results.

Isotherm Models		Freundlich			Dubinin–Radushkevich			
Adsorbates		Parameters			Parameters			
		K_f_	n	R^2^	β	qm (mg/g)	E (KJ/mol)	R^2^
Cu^2+^	Commercial	4.501	0.572	0.9724	0.1549	4.737	1.97	0.941
	Extracted	8.323	0.265	0.9514	0.4024	13.994	1.115	0.9152
Pb^2+^	Commercial	6.482	0.338	0.9842	0.3021	8.696	1.286	0.9853
	Extracted	10.402	0.450	0.9312	0.1935	9.845	1.607	0.9644
MB	Commercial	6.785	0.604	0.9785	0.286	5.847	1.521	0.9985
	Extracted	12.101	0.351	0.9842	0.3581	15.845	1.254	0.9854
CV	Commercial	8.569	0.851	0.9957	0.451	6.487	1.99	0.9965
	Extracted	14.526	0.541	0.9888	0.2547	7.257	1.758	0.9954

**Table 4 materials-17-02724-t004:** Comparison of the results obtained in the present study with similar studies from the literature.

Chitosan from Other Studies				Present Study		
Name	Adsorbate	Adsorption Capacity (%)	References	Name	Adsorbate	Adsorption Capacity (%)
Chitosan	Cu^2+^	9.29	[71]	Chitosan powder	Cu^2+^	98.98
	Pb^2+^	89	[72]	Chitosan cross-linked gel GA	Cu^2+^	85
	CV	8.91	[73]			
Chitosan/poly(vinyl alcohol)	Cu^2+^	13.52	[71]	Chitosan powder	Pb^2+^	80
				Chitosan cross-linked gel GA	Pb^2+^	95
Sodium lignosulfonate/chitosan	Cu^2+^	89.9	[74]	Chitosan powder	MB	98
				Chitosan cross-linked gel GA	MB	85
Chitosan immobilized in alginate beads	Cu^2+^	77	[75]	Chitosan powder	CV	86
				Chitosan cross-linked gel GA	CV	
Chitosan/Acrylic acid	Cu^2+^	88	[75]			
Chitosan-grafted maleic acid	Cu^2+^	88	[76]			
Chitosan/cellulose	Cu^2+^	7.46	[77]			
	Pb^2+^	7.41				
Chitosan/sand	Cu^2+^	13.47	[78]			
	Pb^2+^	3.47				
Glutaraldehyde-cross-linked chitosan	MB	95.22	[79]			
	CV	87				
Cross-linked chitosan/Activated clinoptilolite	MB	40.47	[80]			
Chitosan hydrogel beads	CV	1.54	[81]			
Chitosan–graphite oxide modified polyurethane	CV	18.29	[41]			

## Data Availability

Data are contained within the article.

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
