# Peer review of "Enhancing Wastewater Depollution: Sustainable Biosorption Using Chemically Modified Chitosan Derivatives for Efficient Removal of Heavy Metals and Dyes"

_materials, 2024, doi:10.3390/ma17112724_

Round 1
Reviewer 1 Report (Previous Reviewer 3)
Comments and Suggestions for Authors
Dear Authors,
The topic of your work is interesting and current, however, I have some remarks and suggestions about your work.
Shorten the abstract
Shorten the introduction, be concise.
Your BET curves are not good and you did not give the distribution of pores and alpha s
You presented a lot of results, but it seems that your results are not connected, as if you threw the results in and did some analyzes unnecessarily.
Author Response
please find attached

Reviewer 2 Report (Previous Reviewer 5)
Comments and Suggestions for Authors
The article presented an interesting approach in creating a chemically modified Chitosan(?) that seems to be a cross-linking chemistry. The results are promising. However, the manuscript is needed to be improved before it is ready to be published in Materials. The itemized comments are as follows:
1. What is the term "chemically modified" in this manuscript seems to strictly refer to "chemically crosslink". Please consider adjusting the title accordingly.
2. It would be beneficial to present a scheme summarizing the chemical reaction and resulting chemical structures. This would help reader to visualize and understand the material structures and properties better.
3. Page 3 Line 33 - What does GA refer to?
4. Page 8 Line 14 - please check the sentence structure - unit is also missing.
5. Figure 4 - misspell "chitosane" - Please also check for errors and typos throughout the manuscript as well.
6. Figure 6 - addition of a rescaled or log plot of Absorption vs time would be easier and more beneficial to demonstrate the decontamination performance/trend at a lower time point. Please consider adding graphs at lower time point in the supplementary.
7. What is the deacetylation degree (DD) of commercially available chitosan? Since the manuscript showed 83.88% deacetylation with a drastic difference in performance, knowing the DD of its counterpart would be more beneficial for readers.
Author Response
please find attached

Reviewer 3 Report (New Reviewer)
Comments and Suggestions for Authors
In this manuscript, the adsorption of lead, copper and two dyes (Methylene Blue and Crystal Violet) was investigated using chitosan powder and hydrogel as an adsorbents. The effect of different parameters as well as isotherm studies were performed. The topic of water pollutants removal is undoubtedly relevant, however, this work will not contribute new knowledge to the field.
The adsorbents investigated are not new. Both chitosan and cross-linked chitosan (not only with glutaraldehyde) have been the subject of research for many years now and are currently of little interest. The chemical method of obtaining chitin is well known, and if the authors presented new elements of this method, they are not sufficiently highlighted and described. In addition, at such low concentrations of contaminants, the adsorbents do not show high efficiency. The comparison of removal of metals and dyes with other adsorbents is very imprecise, e.g. the authors compare the percentage removal at much higher initial concentrations than in this paper, giving a misleading impression of much higher efficiency (Kuczajowska-Zadrożna, Filipkowska, and Jóźwiak 2020; Wan et al. 2010).
The paper is too long and difficult to read, and many elements are not prepared according to the instructions for authors.
In my opinion, this paper is not suitable for publication in Materials.
Author Response
please find attached

This manuscript is a resubmission of an earlier submission. The following is a list of the peer review reports and author responses from that submission.
Round 1
Reviewer 1 Report
Comments and Suggestions for Authors
Previous observations and updates
Abstract.
1. What type of pollution? The idea is very generic and is not specific to the work.
2. Which chitosan derivatives? Specify (adaptation made)
3. Specify which chemical modifications (adaptation made)
4. The comma after "Scanning Electron Microscopy (SEM)," is not necessary and can be omitted to make the text flow better. (adaptation made)
5. Improve writing, for example: The adsorption capacity of chitosan hydrogel cross-linked with glutaraldehyde agent (GA) for Cu(II), Pb(II), MB, and CV was as follows: 85%, 95%, 85%, and 98%, respectively (adaptation made)
6. …to be exceptionally high (approximatively 98%)…. Approximately (adaptation made)
Introduction.
…Water pollution is a significant issue affecting people and nature, primarily due to industrialization and globalization. The situation is projected to become worse in the upcoming years (References). (adaptation made)
Different contaminants, including organic and inorganic substances, are present in water, leading to environmental harm and have been proven to be harmful and cancer-causing… (Very generic information without references). (adaptation made)
2.2. Shrimp shells preparation
The shrimp shells underwent a meticulous cleaning process to eliminate any dirt or impurities. They were thoroughly washed multiple times with tap water to remove salts, sands, and other undesired substances. Following the washing procedure, the shells were further rinsed with distilled water to ensure an even higher level of cleanliness…. (repetitive text, concentrate the washing procedure) (adaptation made)
…The resulting chitosan was subjected to a drying process in an oven for a few hours to remove any remaining moisture…(Mention time and temperature used) (adaptation made)
2.3. Adsorption experiments of metal ions
In equation (1), place 0 and f as subscripts
2.5.4. Swelling study.
In equation (3), place S and 0 as subscripts
(adaptation made)
Abstract:
Lines 23 and 28. Repetitive use of "promising approach"
2.1.2. Chitosan extraction steps
P. 5 Line 5. Change the word “eight” for the number “8”
Error! Reference source not found.
P.7 Line 12, 21, 22, 23, 24, 46
P.8 Line 1, 11
P.9 line 9
P.10 Lines 5, 7, 16, 29
P.13 Line 18
P. 12 Line 6
P. 15 line 52
P. 16 line 25
***In the opinion of the editors, is it possible to arrange table 3 on the sheet horizontally?
A comparison of the chitosan removal capacity between previous studies and our experimental results
P. 16 line 31 double “the the”
Although the text is clear and objective, there are multiple ideas that are repetitive on several occasions. concentrate the main ideas and reduce the text.
Comments on the Quality of English LanguageMinor editing of English language required
Reviewer 2 Report
Comments and Suggestions for Authors
1. Abstract should be more concise to state briefly the purpose of the research, the principal results and major conclusions.
2. Introduction also should be more concise to state the developmemt and the problems using Chitosan-based powder and hydrogel for the adsorption of pollutants from aqueous solution, and the first three paragraphs should be deleted.
3. What are the novelty ans advantages of this manuscript compared with the reported similar ones?
4. Table 3 should be optimized to well present the adsorption properties comparsion of the as-prepared adsorbents with the reported ones.
5. Conclusions should be more concise to state the main results of this study.
6. The language should be polished.
Comments on the Quality of English LanguageThe language should be polished and more concise.
Reviewer 3 Report
Comments and Suggestions for Authors
Dear All,
You presented a lot of random results, I didn't see any innovation in your research
The title of the work is unclear, besides, what does Ideal Chitosan Powder mean?
Shorten the abstract, it looks like an introduction... Be concise and apostrophize only important things
Also, the introduction is too long, shorten it and remove unnecessary things
Why didn't you do a proximate and ultimate analysis of the scampi shell
Show a picture of the shell of the scampi, after washing and when you have obtained a granulation of 1 mm
Explain section 27 ? lyophilization is something else
How did you grind the sample, with what, did you use sieves?
Paragraph 3.1 sentence,,Error! Reference source not found,, I don't know where you saw that someone puts this into operation and repeats it several times
Picture 1. Technically arrange, separate
Table 1. How did you manage to get 0.27? And if it is, I don't see the purpose of that data in the paper.
Be sure to show me the isotherms and other graphics and diagrams that go with the BET method
I have not seen anywhere that you have given which devices BET, XRD, SEM, NMR and other methods were used for
The results are thrown at you, the discussion between the analyzes is not well connected.
Also the additional material is huge for you even though it is additional material.
Reviewer 4 Report
Comments and Suggestions for Authors
The authors study the properties of commercial chitosan and chitosan extracted by the authors as well as a cross-linked chitosan hydrogels with respect to the adsorption of heavy metal ions and dyes. The research design is not novel or creative, but mostly sound. However, there are many errors in the manuscript, it lacks a comprehensive discussion and some measurement results are missing. Thus, I recommend to reject the manuscript in the present form.
Detailed comments:
Abstract: The abstract needs to be rewritten. There are several sentences, which are incomplete or exhibit other major grammar/language errors. Furthermore, the abstract is to lengthy with too many details, but too little emphasis on the larger picture of the study.
Page 1
Lines 19/20: There is something amiss with the sentence at the end. Please fix. There are also multiple English language errors in the following sentences in the abstract. Please fix those, too.
Line 22/23: How was it modified?
Line 43: Dubinin?
Page 2
Line 14: “in this field.” Please specify
Page 3
Line 23: Full stop at the end of the sentence is missing.
Page 4
Please specify the suppliers of all chemicals used.
Lines 27-28: I highly doubt that the shells were turned into a gas as stated. I believe the authors mean that remaining water was removed from the shells by lyophilization (better known as freeze drying). Please correct this.
In the explanation about the extraction of the chitosan, the authors switch from the word chitin to the word chitosan at an out of place position in the text. It seems as if the authors did not understand which step turns chitin into chitosan. The authors should educate themselves and the readers about the deacetylation process.
There are numerous small errors throughout the manuscript, e.g. erroneous capitalization, misuse of punctuation, missing words etc., which should be fixed.
Page 7
All the references to figures went missing, starting from page 7. Please correct this.
Line 14: The authors state, that the average particle size is 255.61 µm. I doubt that this accuracy is meaningful. What is the error of this method?
Line 31: XRD does not result in a “spectrum”. Please educate yourselves on this topic and use a correct term.
Line 31: Please provide the XRD data of the cross-linked gel.
Line 47: Please provide the isotherms and pore size distributions obtained by N2 physisorption. What could be the potential cause for this very different surface area of commercial and extracted chitosan? If there are mesopores, this should be visible in the pore size distribution.
Figure 1: The method of “granulometry” is used. It should be explained in the methods section of the supporting information.
Table 1: It should be discussed, why there is such a big discrepancy between the nitrogen content in commercial and extracted chitosan (factor 2!). It should also be discussed, why the results from XPS do not match the theoretical composition of chitosan.
Figure 7: Please keep the color of a certain sample the same in every graph.
Figure 7: It seems odd, that the adsorption capability should not correlate with the surface area (factor 27!!). Furthermore, the impact of the
Page 15: Discussion of adsorption isotherm models: By looking at graphs S8-S11, it does not become clear why one adsorption model should fit better then another. The plots all look pretty much the same. A more detailed discussion/explanation is needed here.
Supporting Information:
Please improve formatting of symbols especially in equations and graphs and make sure to explain the abbreviations used.
Please show figures in the same order as mentioned in the main manuscript.
Figures S9 and S10 have the same caption!
Comments on the Quality of English LanguageThe authors study the properties of commercial chitosan and chitosan extracted by the authors as well as a cross-linked chitosan hydrogels with respect to the adsorption of heavy metal ions and dyes. The research design is not novel or creative, but mostly sound. However, there are many errors in the manuscript, it lacks a comprehensive discussion and some measurement results are missing. Thus, I recommend to reject the manuscript in the present form.
Detailed comments:
Abstract: The abstract needs to be rewritten. There are several sentences, which are incomplete or exhibit other major grammar/language errors. Furthermore, the abstract is to lengthy with too many details, but too little emphasis on the larger picture of the study.
Page 1
Lines 19/20: There is something amiss with the sentence at the end. Please fix. There are also multiple English language errors in the following sentences in the abstract. Please fix those, too.
Line 22/23: How was it modified?
Line 43: Dubinin?
Page 2
Line 14: “in this field.” Please specify
Page 3
Line 23: Full stop at the end of the sentence is missing.
Page 4
Please specify the suppliers of all chemicals used.
Lines 27-28: I highly doubt that the shells were turned into a gas as stated. I believe the authors mean that remaining water was removed from the shells by lyophilization (better known as freeze drying). Please correct this.
In the explanation about the extraction of the chitosan, the authors switch from the word chitin to the word chitosan at an out of place position in the text. It seems as if the authors did not understand which step turns chitin into chitosan. The authors should educate themselves and the readers about the deacetylation process.
There are numerous small errors throughout the manuscript, e.g. erroneous capitalization, misuse of punctuation, missing words etc., which should be fixed.
Page 7
All the references to figures went missing, starting from page 7. Please correct this.
Line 14: The authors state, that the average particle size is 255.61 µm. I doubt that this accuracy is meaningful. What is the error of this method?
Line 31: XRD does not result in a “spectrum”. Please educate yourselves on this topic and use a correct term.
Line 31: Please provide the XRD data of the cross-linked gel.
Line 47: Please provide the isotherms and pore size distributions obtained by N2 physisorption. What could be the potential cause for this very different surface area of commercial and extracted chitosan? If there are mesopores, this should be visible in the pore size distribution.
Figure 1: The method of “granulometry” is used. It should be explained in the methods section of the supporting information.
Table 1: It should be discussed, why there is such a big discrepancy between the nitrogen content in commercial and extracted chitosan (factor 2!). It should also be discussed, why the results from XPS do not match the theoretical composition of chitosan.
Figure 7: Please keep the color of a certain sample the same in every graph.
Figure 7: It seems odd, that the adsorption capability should not correlate with the surface area (factor 27!!). Furthermore, the impact of the
Page 15: Discussion of adsorption isotherm models: By looking at graphs S8-S11, it does not become clear why one adsorption model should fit better then another. The plots all look pretty much the same. A more detailed discussion/explanation is needed here.
Supporting Information:
Please improve formatting of symbols especially in equations and graphs and make sure to explain the abbreviations used.
Please show figures in the same order as mentioned in the main manuscript.
Figures S9 and S10 have the same caption!
Reviewer 5 Report
Comments and Suggestions for Authors
The manuscript titled "Synthesis and Optimization of Ideal Chitosan Powder and Hydrogel from Shrimp Shells for Biofiltration System" focuses on demonstrating the result of optimization of several factors of chitosan film formation to maximize the absorption capacity of the films for filtration system. The manuscript presents many interesting results which makes the manuscript sorely need a better explanation about the synthesis method and the mechanism behind such a high efficiency chitosan film. Aside from the obvious “Error! Reference source not found” issues for citations, figures, and tables, other issues must be addressed before the manuscript is suitable to be reevaluate and consider to be published in Materials. Some itemized issues are listed below:
1. Figure 1 - The results of particle size distribution and BET are not well correlated, since the commercial got more volume (%) than the extracted one why is that the case?
2. Figure 1 - The XRD spectra of the extracted chitosan show peaks at around 30 degrees that b1 doesn’t have. What is that peak? What the significance it has for the material?
3. Table 1 - the result of BET surface area between the commercial and extracted very different which is to be expected from lyophilization. However, the BET suggested minimal difference between the two. Provide some explanation and photo comparison between the two samples.
4. Figure 2. - Please color code your FTIR graph so it is conformed with all your color coding from other Figures.
5.Figure 5. - pH of 7.2 would be considered very mildly basic and should be mentioned as such. A clearer comparison between acidic/neutral/basic pH condition might be required here.
6. Figure 7. - For the sake of consistency, the authors should use the same color-coding scheme across all Figures.
7. The abbreviation is not intuitive please consider renaming them as it can confuse the reader.
8. Table3 - There is no data on Chitosan cross-linked gel GA of CV
9. Why Cu2+ and Pb2+ results between powder and gel are seemingly swapped? If this is not a mistake, please provide some detailed explanation.
10. Compared to the previous study, this study shows highly efficient adsorption capacity on chitosan material. This warrants a more in detailed explanation and comparison - not in the term of result but in the term of material properties. This must be provided before further consideration can be made.
11. The authors should show the result with control to compare, also as commercial, extracted chitosan and hydrogel together. This applies for both table 2 and 3.
12. The authors should redo table 3 as it is hard to follow in the present form.
13. On the the effect of contact time, pH, temperature, and concentration, the authors should comply and show the results in a table in the main text for ease of comparison (The graphs in the supplementary file are fine - though some errors in writings could be spotted).
14. However, the results in the supplementary are vary significantly from one another for the metal ions (2+) and dyes (similar chemical structure) that are tested, explain the interaction of the extracted chitosan on each item in more detail.
15. If one subjected the commercially available chitosan through the same lyophilization method, will the absorption results differ than starting from scratch like this manuscript does?
16. The conclusion does not provide the optimized parameter of the chitosan for the proposed application. Please be more specific about the optimization and present the finding accordingly since the tittle of the manuscript claims so.